# Organocatalytic diastereo- and atroposelective construction of N−N axially chiral pyrroles and indoles

**Shao-Jie Wang[1,2], Xia Wang[1], Xiaolan Xin[1], Shulei Zhang[1], Hui Yang [2],
Ming Wah Wong [2] ✉ & Shenci Lu [1] ✉**

The construction of N−N axially chiral motifs is an important research topic, owing to their wide occurrence in natural products, pharmaceuticals and chiral ligands. One efficient method is the atroposelective dihydropyrimidin-4-one formation. We present herein a direct catalytic synthesis of N−N atropisomers with simultaneous creation of contiguous axial and central chirality by oxidative NHC (*N*-heterocyclic carbenes) catalyzed (3 + 3) cycloaddition. Using our method, we are able to synthesize structurally diverse N−N axially chiral pyrroles and indoles with vicinal central chirality or bearing a 2,3-dihydropyrimidin-4-one moiety in moderate to good yields and excellent enantioselectivities. Further synthetic transformations of the obtained axially chiral pyrroles and indoles derivative products are demonstrated. The reaction mechanism and the origin of enantioselectivity are understood through DFT calculations.

Recently, the wide applications of axially chiral biaryls in catalysis and drug delivery have excited much research interest in their synthesis[1,2]. Among them, atropisomeric indole derivatives are particularly important as synthetic precursors of many pharmaceuticals[3,4]. The last decade has witnessed many excellent research works in the construction of axially chiral indole-based frameworks[5]. Nitrogen-nitrogen atropisomers, an important subclass arising from sterically hindered rotation around a single N−N bond, are of high natural abundance[6–9], as exemplified by Dixiamycin A and Schischkiniin. In addition, 2,20-bis(diphenylphosphino)-1,10-bibenzimidazole (BIMIP) was reported as a diphosphine ligand (Fig. 1a)[10,11]. Irrespective of their significant utility, effective enantioselective synthesis of N−N atropisomers was not explored until recently, Liu[12–15], Sparr[16], You[17], Sun[18], Li[19], Zhao[20], Shi[21,22], Lu and Houk[23], and Li[24,25] reported several methods for the synthesis of N−N bispyrrole, indole pyrrole, bisindole, and nonbiaryl atropisomers via asymmetric copper-catalysis, palladium-catalysis, iridium-catalysis, rhodium-catalysis, chiral phosphoric acid catalysis, or Brønsted base (Fig. 1b). However, these studies mainly focused on the atroposelective construction of N−N axial chirality. Besides, the highly atroposelective

creation of N−N axially chiral pyrrole/indole-based heterocyclic six-membered ring remains unknown[19]. Recently, simultaneously controlling multiple chiral elements (axial chirality and central chirality) has emerged as an important research area with several pioneering works reported in the past decade[26–28]. Despite of the excellent progress made by these groups, it remains very challenging to construct contiguous axial and central chirality in a high diastereo- and enantioselective manner, including (1) efficient cyclization between two sterically demanding partners, (2) the control of enantioselectivity, and (3) creation of vicinal axial and central chirality in a single operation with potential diastereoselectivity issues. Very recently, elegant catalytic enantioselective syntheses of atropisomeric hydrazides were reported, through a one-pot sequence of two organocatalytic cycles[29]. However, the direct catalytic synthesis of N−N atropisomers with simultaneous creation of contiguous axial and central chirality remains elusive in the literature, which represented a significant gap in synthetic method development.

N-heterocyclic carbene (NHC) catalysis has emerged as a powerful tool in the preparation of both central and atropisomeric chiral

[1]Frontiers Science Center for Flexible Electronics (FSCFE), Shaanxi Institute of Flexible Electronics (SIFE) & Shaanxi Institute of Biomedical Materials and Engineering (SIBME), Northwestern Polytechnical University (NPU), 127 West Youyi Road, Xi'an 710072, China. [2]Department of Chemistry, National University of Singapore, 3 Science Drive 3, Singapore 117543, Singapore. ✉e-mail: chmwmw@nus.edu.sg; iamsclu@nwpu.edu.cn

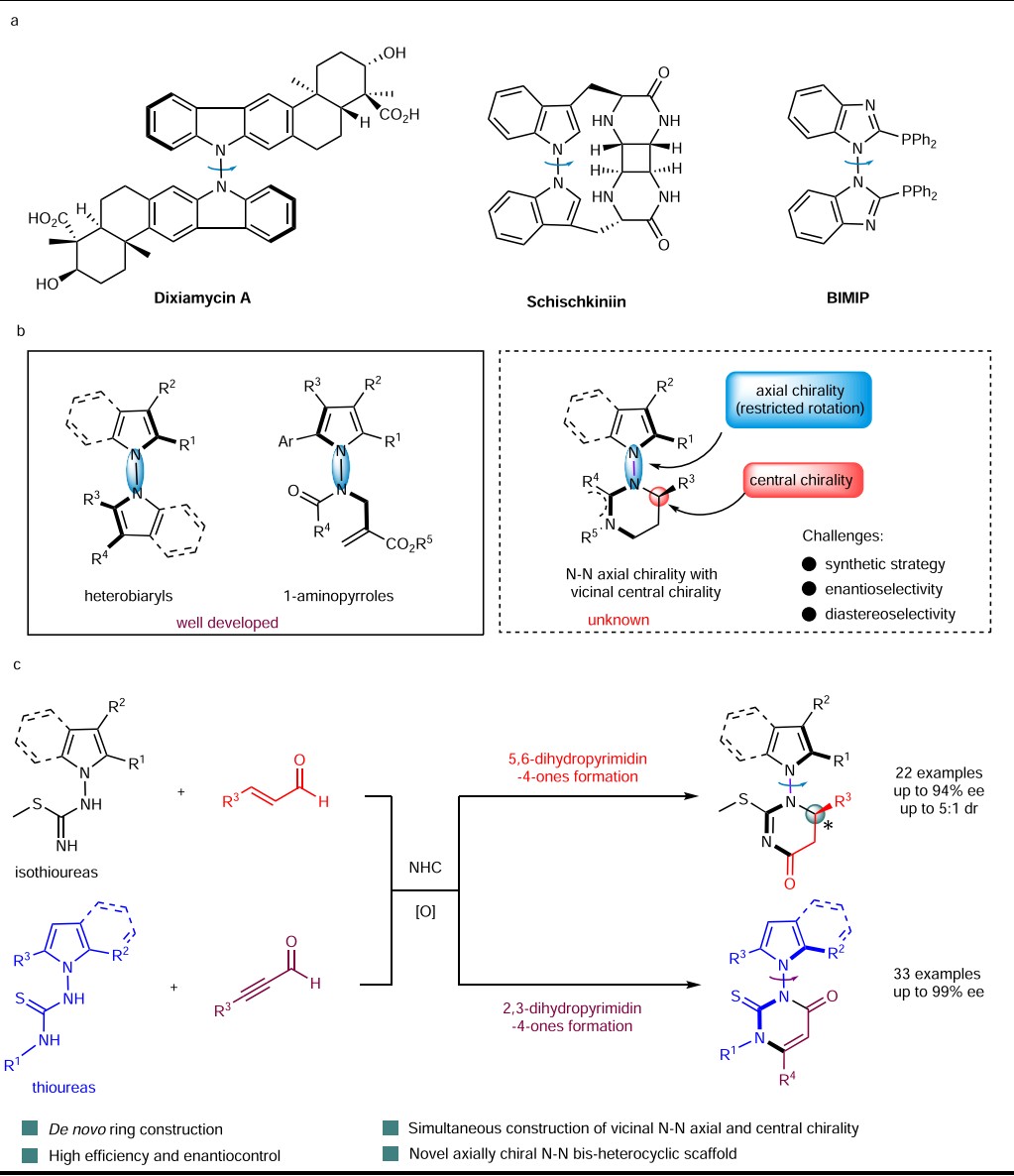

**Fig. 1 | Asymmetric catalytic synthesis of N-N atropisomers. a** Important N-N atropisomers wtih bis-heterocyclic skeletons. **b** Recent examples of N-N atropisomers bearing pyrrole and/or indole scaffolds. **c** This work: synthesis of N-N atropisomers wtih bis-heterocyclic skeletons by oxidative NHC catalysis.

molecules[30–50] and in the construction of six-membered rings through asymmetric (3 + 3) cycloaddition using α,β-unsaturated acylazoliums as C3 synthon[36]. To the best of our knowledge, no example of the synthesis of N–N axially chiral pyrroles and indoles catalyzed by N-heterocyclic carbene has been reported[51]. We are interested in NHC-catalyzed asymmetric reactions[52–58]. Here, we show an efficient access to diastereo- and atroposelective N–N axially chiral pyrroles and indoles with vicinal central chirality through a highly atroposelective (3 + 3) cycloaddition between readily available isothioureas with enals via oxidative NHC catalysis. Besides, this strategy also enabled facile access to N–N axially chiral pyrroles and indoles with excellent atroposelectivity starting from readily available thioureas with ynals through oxidative NHC catalysis (Fig. 1c).

## Results
### Diastereo- and atroposelective synthesis of N–N axially chiral indoles/pyrroles with vicinal central chirality
Initially, we designed the oxidative NHC catalytic synthesis of 5,6-dihydropyrimidin-4-ones from readily available isothiourea **1a** and

cinnamaldehyde **2a** with central chirality followed by central-to-axial chirality conversion to produce N−N atropisomers with bis-heterocyclic skeletones (Fig. 2a). The Chi's group reported an NHC-catalyzed addition of isothioureas to enals under oxidative conditions to form 5,6-dihydropyrimidin-4-ones bearing stereocenters with high optical purity[59]. When we carried out the reaction of **1a** with cinnamaldehyde **2a** in the presence of azolium catalyst, oxidant, NaOAc, and HOAc, to our great surprise, two diastereomeric compounds **3a** and **3a'** (about 1.5:1 dr), could be easily separated by column chromatography; both isomers were obtained with high ee of 86% (Fig. 2b). The absolute configuration of **3a** and **3a'** were unambiguously assigned by single crystal X-ray diffraction analysis, respectively. The configuration of the other products was assigned by analogy. Attracted by this unprecedented one-pot catalytic synthesis of N−N axially chiral indoles with vicinal central chirality, we turned our attention to the exploration of this intriguing transformation.

Our next goal is to optimize the reaction to improve both enantio- and diastereoselectivity. After a brief survey of NHC precatalysts, the aminoindanol-derived NHC generated from the chiral triazolium salt **F**

**Fig. 2 | Reaction design and discovery. a** Our design: construction of central chirality followed by central-to-axial chirality conversion. **b** Beyond design: synergistic construction of central and axial chirality via a single stereodetermining steps.

bearing N-2,4,6-triisopropylphenyl group was found to the best catalyst. In the presence of NaOAc as the base, the major diastereomer **3a** could be obtained in 68% NMR yield with 92% ee as well as the minor diastereomer **3a'** in 23% NMR yield at 30 °C in toluene (Table 1, entry 6 vs entries 1–5). The solvent screening revealed that DCM, ethyl acetate (EA), and THF resulted in reduced selectivity and yields (Table 1, entries 7 – 9). An extensive base screening revealed that bases such as DBU could not furnish the desired product (Table 1, entry 10), whereas DIPEA, KO^tBu, and KOAc could afford the desired product in reduced selectivity and yield as compared to NaOAc (Table 1, entry 6 vs entries 11 – 13). Lastly, other additives were tested such as 3 Å MS, 5 Å MS, Na₂SO₄, and MgSO₄. It was shown that the addition of MgSO₄ could help to afford the **3a** with the highest yield and excellent ee.

We then proceeded to examine the generality of this catalytic (3 + 3) cycloaddition reaction using the optimized reaction conditions (Fig. 3). A range of enals were examined as substrates using isothiourea **1a** as the complementary reactant to furnish products **3b-3m**. Various enals containing electron-donating and electron-withdrawing substituents at the para position of the β-aryl moiety were shown to be effectively converted under these developed conditions. In all cases examined, the indole-derived 5,6-dihydropyrimidin-4-one products were obtained in good yields and enantiomeric excess values (**3b-3h**). Furthermore, enals possessing meta-substituents were shown to undergo smooth (3 + 3) cycloaddition, furnishing the desired products in moderate to good yield with excellent stereoselectivity (**3i-3j**). Additionally, substitution at the β-position of the enal with a heteroaryl moiety did not impact the outcome of the reaction, and the corresponding furyl adduct **3k** was synthesized successfully. Moreover, this methodology was found to be extensible to β-vinyl enals, generating

the desired cycloadduct (**3 l**) in moderate yield and appreciable enantioselectivity. Additionally, alkyl enal such as hex-2-enal were amenable to the developed conditions, furnishing two diastereomeric products (**3m**) and (**3m'**) that could be readily isolated in 73-80% ee following flash column chromatography.

Next, the variations on isothioureas were studied. The substrate with different substitutions on the phenyl moiety of isothioureas as well 2,3-dimethyl or cyclohexyl groups also gave the corresponding products (**3n-3r**) in moderate to high yields and excellent ee. To further expand the scope of the substrate, the pyrrole based isothiourea was also tested. Electron-withdrawing, and electron-donating groups at the para-position of cinnamaldehyde as well 3-(furan-2-yl)acrylaldehyde were well tolerated to afford the corresponding products **3s-3v** in good yield and enantioselectivity.

The two rotamers **3a** and **3a'** were oxidized by Iodine in DMSO at room temperature to furnish atropisomeric indole-based pyrimidin-4-one **4** and its enantiomer *ent*-**4**, respectively, without any loss of enantiomeric excess (Fig. 4a, b). Simultaneously, iodo substituent was introduced onto the 3-position of indole, which served as a potential transformation handle. The absolute configuration of **4** was unambiguously assigned by single crystal X-ray diffraction analysis.

## Atroposelective synthesis of N−N axially chiral pyrroles and indoles

Besides, alkenyl acylazoliums, another important type of NHC-bound intermediates such as the alkynyl acylazoliums were also investigated as a C3 synthon. Our next goal is to create other type of N−N axially chiral pyrroles or indoles using (3 + 3) cycloaddition catalyzed through alkynyl acylazoliums (II)[60–63]. Initially, the isothiourea **1a** was chosen as

**Table 1 | Optimization of the reaction conditions[a]**

| Entry | preNHC | Solvent | Base | Additive | dr[b] | 3a, yield (%)[c] | 3a, ee (%)[d] |
|---|---|---|---|---|---|---|---|
| 1 | A | toluene | NaOAc | 4 Å MS | 1.2:1 | 22 | 49 |
| 2 | B | toluene | NaOAc | 4 Å MS | 1.4:1 | 30 | 51 |
| 3 | C | toluene | NaOAc | 4 Å MS | 1:1 | 20 | 55 |
| 4 | D | toluene | NaOAc | 4 Å MS | 3:1 | 25 | 86 |
| 5 | E | toluene | NaOAc | 4 Å MS | 2:1 | 42 | 62 |
| 6 | F | toluene | NaOAc | 4 Å MS | 3:1 | 68 | 92 |
| 7 | F | $CH_2Cl_2$ | NaOAc | 4 Å MS | 2:1 | 33 | 70 |
| 8 | F | EtOAc | NaOAc | 4 Å MS | 2:1 | 40 | 81 |
| 9 | F | THF | NaOAc | 4 Å MS | 3:1 | 58 | 91 |
| 10 | F | toluene | DBU | 4 Å MS | / | trace | / |
| 11 | F | toluene | DIPEA | 4 Å MS | 2:1 | 13 | 66 |
| 12 | F | toluene | KOtBu | 4 Å MS | 2:1 | 44 | 56 |
| 13 | F | toluene | KOAc | 4 Å MS | 3:1 | 60 | 90 |
| 14 | F | toluene | NaOAc | 3 Å MS | 2:1 | 58 | 88 |
| 15 | F | toluene | NaOAc | 5 Å MS | 3:1 | 66 | 89 |
| 16 | F | toluene | NaOAc | $Na_2SO_4$ | 4:1 | 72 | 94 |
| 17 | F | toluene | NaOAc | $MgSO_4$ | 4:1 | 76 (72)[e] | 94 |

[a]Unless noted otherwise, the reactions were performed with **1a** (0.05 mmol, 1.0 equiv.), cinnamaldehyde **2a** (0.09 mmol, 1.8 equiv.), preNHC (20 mol%), **DQ** (125 mol%), AcOH (30 mol%), additive (50 mg) and base (150 mol%) in solvent (1 mL) under $N_2$ atmosphere at 30 °C for 48 h.
[b]The dr (the ratio of **3a** to **3a'**) was determined by [1]HNMR analysis with 1,1,2,2-tetrachloroethane as the internal standard.
[c]The yield of major diastereomer **3a** was determined by [1]H NMR using 1,1,2,2-tetrachloroethane as the internal standard.
[d]Determined by chiral HPLC analysis.
[e]Isolated yield of **3a** with respect to **1a**. Mes = 2,4,6-trimethylphenyl.

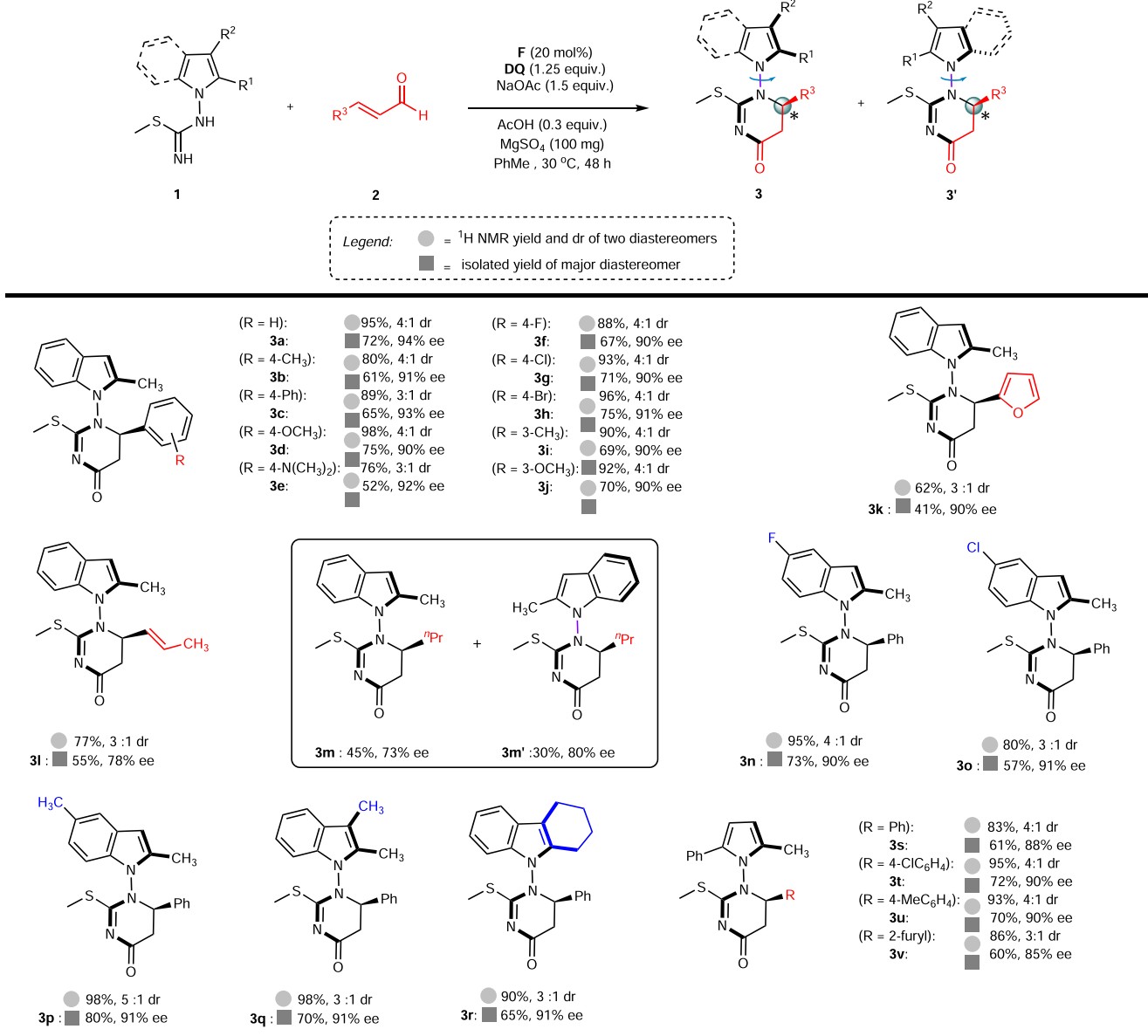

**Fig. 3 | Synthesis of 3 or 3a'.** For details, please see Supplementary Information (SI).

the model substrate, no desired product was obtained after a lot of conditions screening. Later, we switched the substrate isothiourea to thiourea. Fortunately, the N–N axially chiral pyrrole bearing a 2,3-dihydropyrimidin-4-one moiety **7a** was obtained in 36% yield and −77% ee (Fig. 4c). Nevertheless, NHC optimization revealed that the yield and ee of product **7a** was improved to 51% and 97%, respectively. It is noteworthy that no axially chiral product **8** was detected and this reaction proceeded in a chemoselective fashion. The representative optimization studies are included in the Supplementary Information (Supplementary Table 1).

Initially, the commonly used strong electron withdrawing group (CF₃) on the N-phenyl group of the substrate **5** could be well-tolerated to produce (**7b-7c**) with moderate yields and moderate to excellent enantioselectivities of 90–94% ee (Fig. 5). Unfortunately, the substrate bearing a neutral or electron donating group on the N-phenyl group led to complex reaction mixtures in this NHC catalytic conditions. With the identified optimal conditions in hand, the variations of either electron-donating or withdrawing group on para-position of phenyl

moiety of **5** were examined and they were well-tolerated to yield the corresponding products **7d–7j** in moderate yields and high to excellent enantioselectivities (85–93% ee). The different substituents on the meta or ortho position of phenyl ring were also tested, and the corresponding products **7k-7o** were afforded with moderate yields and excellent enantioselectivities (90–99% ee). The absolute configuration of **7l** was unambiguously assigned by single crystal X-ray diffraction analysis. The configuration of the other products was assigned by analogy.

Then we investigated the different substituents on the phenyl ring of the ynals in the reaction. The electronic effect showed obvious influence on the reaction outcome and the corresponding products **7p-7ac** were obtained with moderate yields and excellent enantioselectivities (95–98% ee). Lastly, substitution at the β-position of ynals with heteroaryl moiety did not affect the outcome of the reaction and the corresponding thienyl product **7ad** was formed. To further expand the scope of the substrate, the indole based thioureas were also tested. The 2-substituted indole based thioureas (methyl or

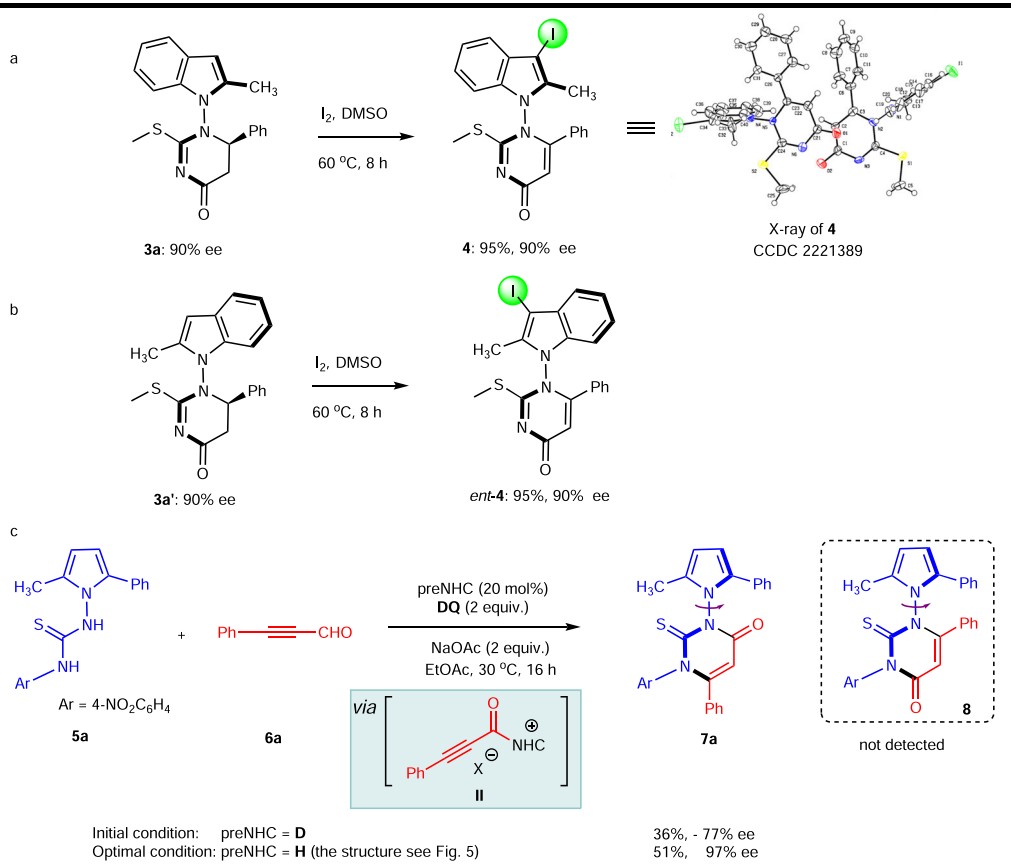

**Fig. 4 | Two types of N-N axis. a** Oxidation of **3a** and **3a'** into enantiomeric compounds **4** and *ent*-**4**. **b** Alternative strategy for synthesizing N−N axially chiral pyrroles. **c** Alternative strategy for synthesizing N−N axially chiral pyrroles.

phenyl group) were well tolerated to afford the corresponding products **7ae**-**7af** in moderate yields and high enantioselectivities. The substrate bearing the useful group such as -P(O)Ph$_2$ successfully provided the desired product **7ag** in 65% yield with 86% ee. When the ortho CF$_3$ group on the N-phenyl group of the substrates **5** were employed to afford the thiazine derivatives with C−N axial chirality (**9a**-**9b**) in moderate yields with moderate enantioselectivities of 54−77% ee. This type of thiazine derivatives with C−N axial chirality was reported by the Jin's group. The ortho isopropyl group on the N-phenyl group of the thioureas were involved in his work[64]. The configuration of **9b** was unambiguously assigned by single crystal X-ray diffraction analysis. The configuration of the other products was assigned by analogy. The symmetrical pyrrole based thiourea was also employed to give the desired product **9c** in 51% yield with 60% ee. The configurational stability of this C−N linked axially chiral compounds was briefly investigated. Experimentally, the ee value of **9c** deteriorated significantly at 100 °C in toluene and became racemic after stirring for 12 h. The barrier to rotation for **9c** was measured at 100 °C in toluene ($\Delta G^{\neq}$ = 29.6 kcal/mol). The details are included in the Supplementary Information (Supplementary Fig. 5).

Next, we investigated experimentally the configurational stability of this type of N−N linked axially chiral compounds. Selected compounds (**3a**, **4**, **7 l**, etc.) were heated in toluene for 72 h at 150 °C. No erosion of enantiopurity was observed, showing that the chiral axes in these compounds are indeed configurationally stable. Our experimental result was further corroborated by the computed rotation energies of the N−N bond (leading to racemization) of **3a**, **4**, and **7 l** (Fig. 6), which were determined to be 37.1, 48.2, and 45.9 kcal/mol at 150 °C (with t$_{1/2}$ of 13 days, >19,000, and 1270 years), respectively[65].

## Transformations and application

A one-mmol-scale reaction was conducted under standard conditions, affording **3a** and **3a´** in 72% and 18% yield, respectively (Fig. 7a). The pyrimidin-4-one product bearing axial with or without vicinal central chirality could be further functionalized via simple operations. The treatment of **3a** with Grignard reagent in THF at −15 °C followed by elimination of H$_2$O during work-up to generate product **10** with 91% yield and 92% ee (Fig. 7b). Additionally, the axially chiral **4** bearing iodo, late-stage coupling with TMS acetylene followed by deprotection can afford the desired product **11** in 95% yield and 91% ee (Fig. 7c). Bromination of axially chiral **7a** with CuBr$_2$ at 40 °C delivered fully-substituted pyrrole **12** in 67% yield and 94% ee (Fig. 7d). The axially chiral phosphine oxide **7ag** was transformed into the phosphine **13** by reduction with HSiCl$_3$ reagent in 84% yield with 85% ee (Fig. 7e)[66]. Notably, compound **13** was used as a chiral ligand in the palladium-catalyzed enantioselective allyl substitution reaction of (*E*)-1,3-diphenylallyl acetate **14** and dimethyl malonate **15**, affording the target product **16** in 81% yield with 40% ee (Fig. 7f). Although the enantioselectivity needs to be improved, this attempt demonstrated the potential application of the constructed axially chiral indole bearing a 2,3-dihydropyrimidin-4-one moiety scaffold for the development of chiral ligand.

## Reaction mechanism and origin of stereoselectivity

Interestingly, when the ee value of **3a** was studied with respect to catalyst enantiopurity, a negative nonlinear effect (NLE) was observed (Fig. 8a), which suggests a second catalyst molecule is involved in the enantio-differentiating transition states, possibly activating the other substrate isothiourea **1a**. Our result is consistent with earlier studies by Huang, and Jin, which also reported a NLE effect in NHC catalysis[67,68]. In

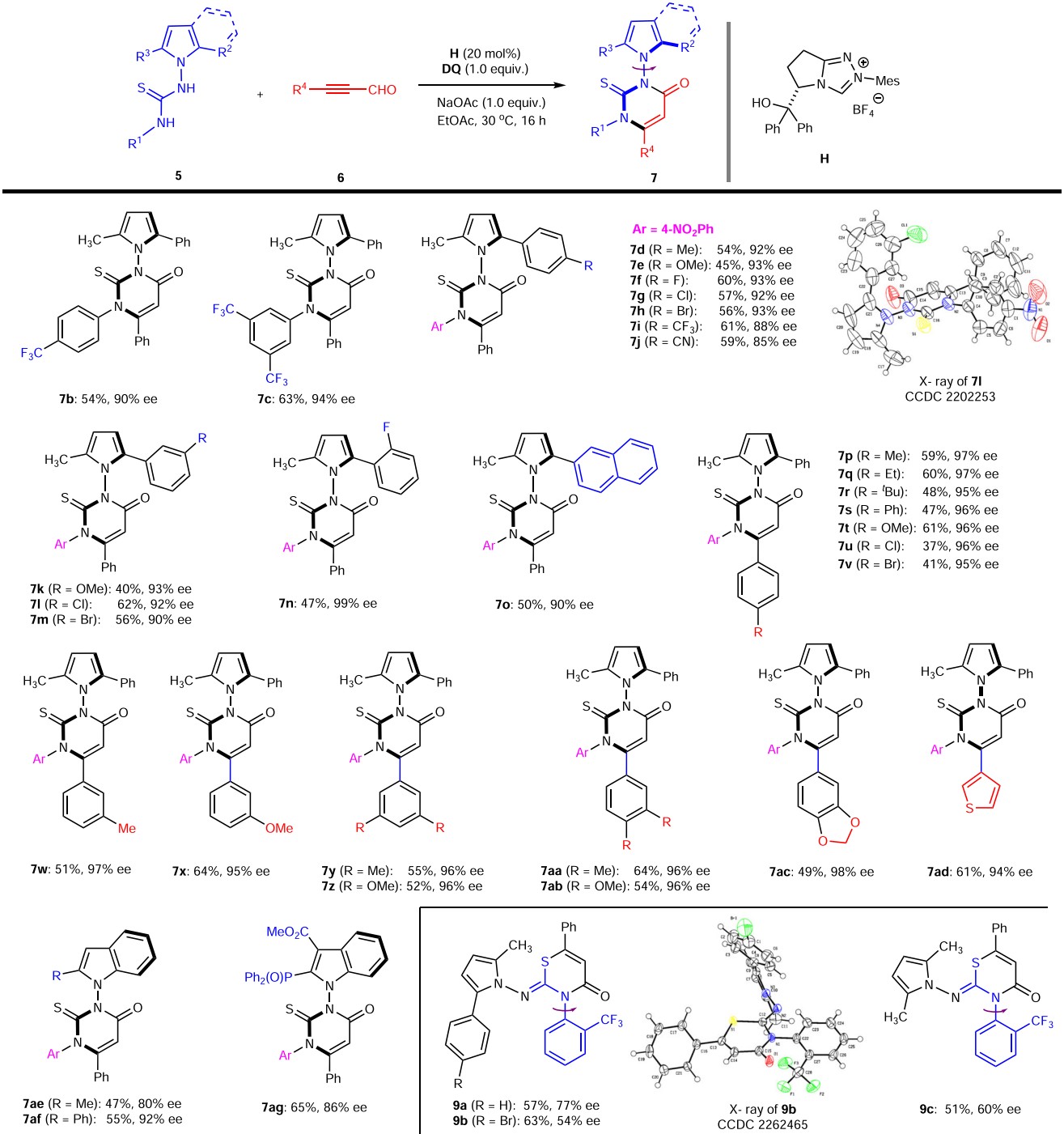

**Fig. 5 | Synthesis of 7 or 9.** For details, please see Supplementary Information (SI).

comparison, a linear effect was observed in NHC pre-catalyst **H** (bearing an OH group) catalyzed synthesis of **7a**, suggesting that in this case, only one catalyst was involved in the enantio-differentiating step (see Supplementary Information for details). One possible explanation for the different behaviors could be that the OH group of pre-catalyst **H** forms a hydrogen bond in the enantio-differentiating step, alleviating the need to involve a second catalyst molecule[37].

To substantiate the proposed interaction between **F** and isothiourea **1a**, we carried out [1]H NMR analysis (Fig. 8b). Upon addition of isothiourea **1a** to a mixture of pre-catalyst **F** and base NaOAc, the chemical shift of the acidic proton of **F** showed a downward shift of 0.15 ppm (Fig. 8b, 1 vs 2). This result is in support of the existence of

interaction between **F** and isothiourea **1a**. Taken together with the NLE experiment, it is possible that the second NHC molecule helps to activate isothiourea **1a** through N-H···C hydrogen bond[55,69–75]. (for more details, see Supplementary Information).

With the above experimental mechanistic insights established, we next proceeded to study the mechanism computationally. Chemistry of NHC catalysis has been well established, and much mechanistic detail prior to the key stereoselective steps, e.g. formation of **F-2** from **F** and **2a**, can be understood without carrying out further experiments or calculations. Based on previous studies and our mechanistic investigations, we initially proposed a catalytic cycle for NHC catalyst **F** as illustrated in Supplementary Figure 350.

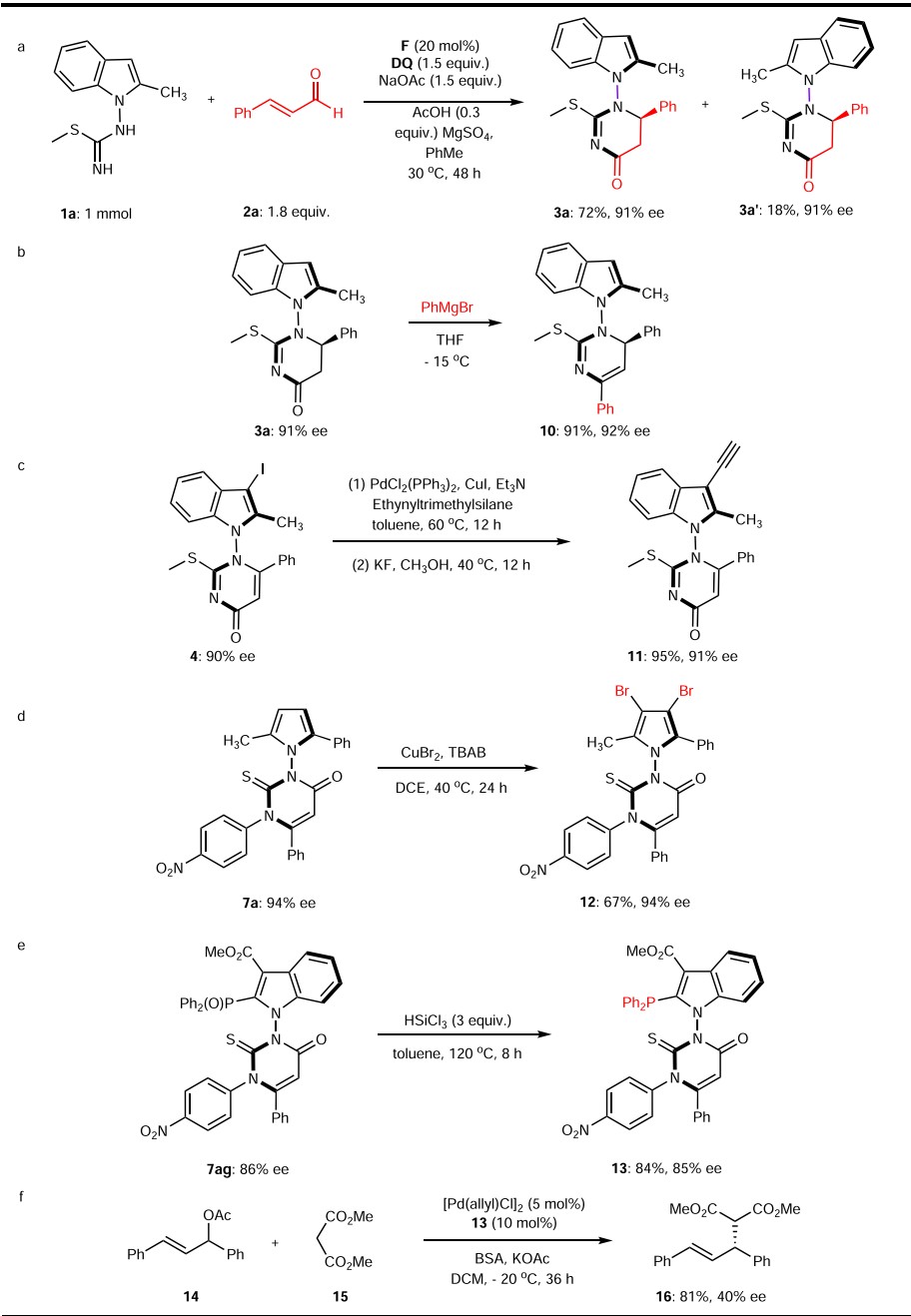

**Fig. 6 | The N–N axis stability investigations.** DFT calculation on the rotation energy of **3a**, **4**, and **7 l**.

**Fig. 7 | Synthetic applications. a** one-mmol-scale reaction. **b–e** Transformations of axial products. **f** Application of chiral phosphine ligand.

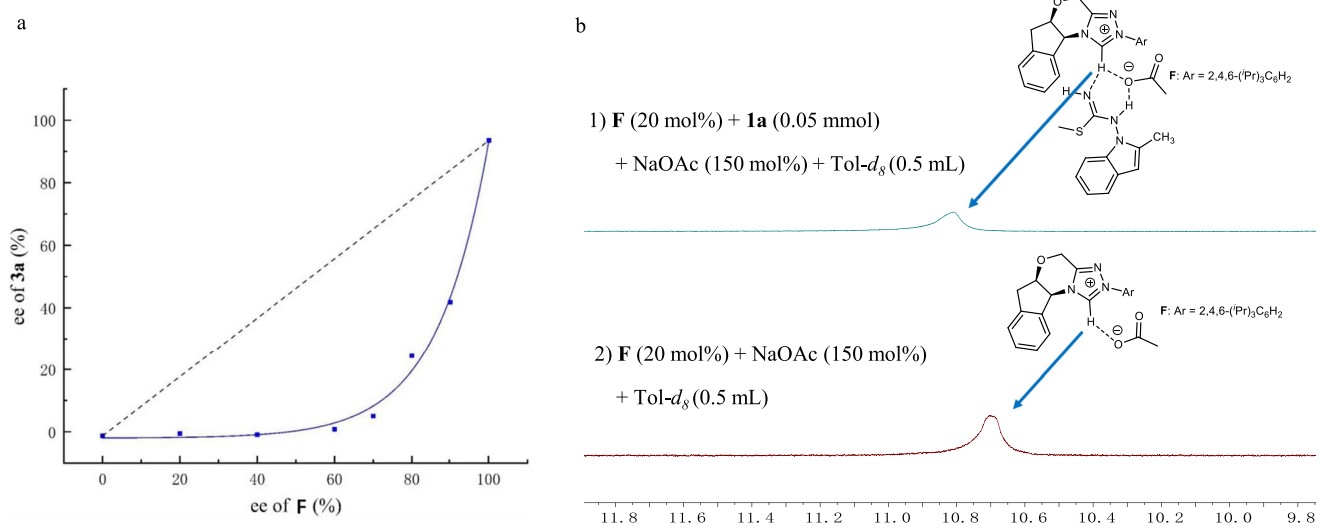

**Fig. 8 | Mechanistic studies. a** Nonlinear effect with respect to product ee and catalyst ee. **b** Chemical shift of the acidic proton of the NHC pre-catalyst **F** under various conditions.

## Discussion

In conclusion, we have developed an oxidative NHC-catalyzed atroposelective synthesis of N–N axially chiral compounds from readily available starting materials. The contiguous N–N axial and central chirality could be simultaneously constructed in a single operation. The reaction took place smoothly under mild conditions and showed good functional group tolerance, which can be allowed for the synthesis of a variety of N–N axially chiral pyrroles and indoles (more than 50 examples) in moderate to good yields, moderate to good diastereoselectivities, and excellent enantioselectivities. Mechanistic studies indicate that non-covalent interactions between the isothiourea and an NHC·HX catalyst play an important role in activating isothiourea towards nucleophilic addition, which was corroborated by DFT calculations. The success of this work not only provides a useful method for the simultaneous construction of contiguous N–N axial and central chirality within a single step but also represents the rare example of catalytic enantioselective synthesis of N–N atropisomers catalyzed by NHC. The development of catalytic procedures to access other challenging structures bearing different types of stereogenic elements is currently under investigation.

## Methods

### General procedure for the Diastereo- and atroposelective synthesis of 3

A dry 4 mL vial with a stir bar was charged with isothioureas **1** (0.1 mmol, 1.0 equiv.), PreNHC **F** (20 mol%), NaOAc (0.15 mmol, 1.5 equiv.), 3,3′,5,5′-Tetra-tert-butyl-4,4′-dibenzoquinone (0.125 mmol, 1.25 equiv.) and MgSO₄ (100 mg). The mixture was taken into the glovebox, where AcOH (0.03 mmol, 0.3 equiv.), enals **2** (0.18 mmol, 1.8 equiv.) and toluene (2.0 mL) were added. The reaction mixture was taken outside the glovebox. The vial was then sealed and the mixture was allowed to stir in the fume hood at 30 °C (oil bath) for 48 h. When the substrate was consumed completely, the mixture was concentrated under vacuum and purified by column chromatography on silica gel (hexane/ethyl acetate = 2:1) to afford the pure product **3**.

### Synthesis of 4

A mixture of **3a** (0.2 mmol, 1.0 equiv.) and iodine (0.4 mmol, 2.0 equiv.) in DMSO (2.0 mL) was warmed at 60 °C in an oil bath for 8 h. On completion of the reaction, the reaction mixture was poured onto a

It was hypothesized that the rotational barrier around the N–N axis in intermediates **F-3** and after is prohibitively high. Therefore, the level of diastereoselectivity is determined by the relative energies of diastereomeric nucleophilic addition transition states from **F-2** to **F-3**. To substantiate this hypothesis, we calculated a model compound truncated from **F-3**, the result of which showed the rotational barrier is very high at 45.3 kcal/mol. At room temperature where the experiments were conducted, such a high barrier will render the axis configurationally stable. Furthermore, N–N axes in compounds similar to **F-3** have also been reported to be configurationally stable by other groups[23].

We investigated computationally the steps starting from the formation of **F-3** from **F-2** of the proposed catalytic cycle. A second carbene molecule was proposed to act as a Brønsted base to activate **1a** through N-H···C hydrogen bond. Our calculated result in Fig. 9 showed the C–C bond cleavage transition state **F-TS6-RR<sub>a</sub>** has the highest energy of 59.4 kJ/mol. The first N–C bond-formation transition state **F-TS1-RR<sub>a</sub>**, the second proton-transfer transition state **F-TS2-RR<sub>a</sub>** and the third ring-closure transition state **F-TS3-RR<sub>a</sub>** have lower energies of 40.6, 53.9, and 6.5 kJ/mol respectively. Barring computational errors, this will imply that the reaction falls within the Curtin-Hammett paradigm and the reaction stereoselectivity will be governed by the C–C bond-cleavage transition states. We noticed that **F-TS6-RR<sub>a</sub>** yields a protonated **3a** as the immediate product, and a proton transfer to **F** is required to regenerate it. We next investigated if the high barrier of this step could be lowered by added base or acid. Indeed, a new reaction pathway with lowered energies was located. Its first step is an acetic acid-catalyzed tautomerization of **F-5** (2.1 kJ/mol) to a lower-energy tautomer **F-8** (−30.9 kJ/mol), which appears to be quite facile. In the presence of the conjugate base sodium acetate, the energy of the C–C bond-cleavage transition state, now as **F-TS5-RR<sub>a</sub>**, could be lowered to 41.7 kJ/mol. Based on the calculated reaction profile and Curtin-Hammett principle, we concluded that the stereochemistry of the reaction is governed by diastereomeric **F-TS2** transition states. However, the presence of many low-lying molecular vibrations (frequencies <30 cm⁻¹), due to the association of a second carbene molecule through a weak hydrogen bond, makes a quantitative prediction of reaction stereoselectivity very challenging. It should be emphasized that our calculations are not in full agreement with the experimental stereoselectivity and there still lacks a clear mechanistic explanation of the negative NLE (for more details, see Supplementary Information).

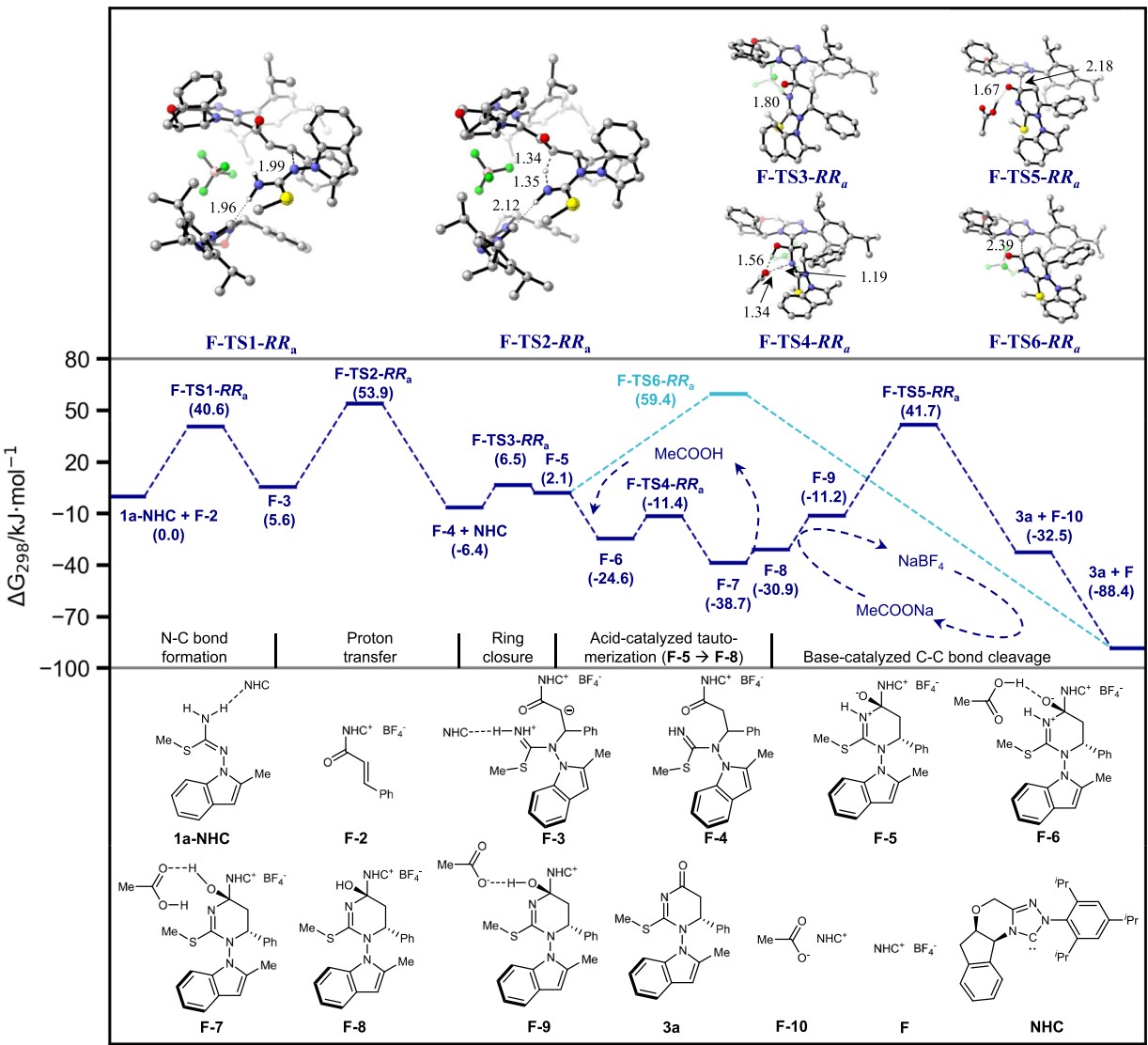

**Fig. 9 | Energy profiles.** Calculated reaction profile at M06-2X/6-311 + G(2d,p)-SMD(toluene)//M06-2X/6-31 G(d)-SMD(toluene) level of NHC-catalyzed atroposelective construction of N–N axially chiral indoles.

saturated solution of sodium thiosulfate. The precipitated solid was collected and the desired product was purified by column chromatography using silica gel with increasing percentage of ethyl acetate in hexane as eluting solvent. The desired product **4** was obtained in 95% yield with 90% ee.

### General procedure for the atroposelective synthesis of 7

To a 4 mL vial was added the thioureas **5** (0.2 mmol, 1.0 equiv.), PreNHC **H** (0.04 mmol, 20 mol%), 3,3′,5,5′-Tetra-tert-butyl-4,4′-dibenzoquinone (0.2 mmol, 1.0 equiv.) and NaOAc (0.2 mmol, 1.0 equiv.). The mixture was taken into the glovebox, where ynals **6** (0.24 mmol, 1.2 equiv.) and EtOAc (2.0 mL) were added. The reaction mixture was taken outside the glovebox. The vial was then sealed and the reaction mixture was allowed to stir in the fume hood at 30 °C (oil bath) for overnight. The crude reaction mixture was directly purified by silica gel column chromatography with hexanes/ethyl acetate (5:1 v/v) as eluent to afford the pure products **7**.

## Data availability

The X-ray crystallographic coordinates for structures reported in this study have been deposited at the Cambridge Crystallographic Data Center (CCDC), under deposition numbers 2206276 (**3a**), 2218428 (**3a′**), 2221389 (**4**), 2202253 (**7 l**) and 2262465 (**9b**). These data can be obtained free of charge from The Cambridge Crystallographic Data Center via www.ccdc.cam.ac.uk/data_request/cif. Data related to materials and methods, optimization of conditions, experimental procedures, mechanistic experiments, and spectra are provided in the Supplementary Information. Coordinates of the optimized structures are provided in the supplementary data file. All data are available from the corresponding authors upon request. Source data are provided with this paper.

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

## Acknowledgements

S.L. is grateful for the generous financial support from the Natural Science Basic Research Plan in Shaanxi Province of China (2020JM-107), the Program for Young Talents of Shaanxi Province (5113200043), the Joint Research Funds of Department of Science & Technology of Shannxi Province and Northwestern Polytechnical University (2020GXLH-Z-023), and the Fundamental Research Funds for the Central Universities. We acknowledge Prof. Yu Zhao (National University of Singapore) for his insightful discussions and generous help with the manuscript preparation.

## Author contributions

S.L. and M.W.W. conceived and designed the study. S.-J.W., X.W., X.X. and S.Z. performed the experiments and prepared the Supplementary Information. H.Y. performed the DFT studies and prepared the Supplementary Information. S.L., H.Y. and M.W.W. wrote the manuscript.

## Competing interests

The authors declare no competing interests.
