## [Peer Review File · Nature Communications]

REVIEWER COMMENTS

Reviewer #1 (Remarks to the Author):

The manuscript by Lu and co-workers reports an oxidative NHC-catalyzed atroposelective synthesis of N–N axially chiral compounds from (iso)thioureas and enals. The yields and the enantioselectivities of reactions are generally good, but the diastereoselectivities are mostly modest. Although the method is somewhat interesting, this work is not suitable for Nat. Commun. due to the reasons elaborated below.

With regard to the novelty of this work: (a) N–N axially chiral chemistry: the organo-catalyzed atroposelective formation of the N–N bond with pyrrole/indole scaffold has been reported by many research groups (refs 9-18). I do not think that the authors make any conceptual contributions in this submission. (b) NHC chemistry: NHC-catalyzed selective addition of isothioureas to enals to access 5,6-dihydropyrimidin-4-ones has been reported by the Chi group, however that pioneering work is not cited in this manuscript (Org. Chem. Front., 2021, 8, 743). NHC-catalyzed atroposelective cycloaddition reaction between thioureas and ynals has also been developed by the Jin group, however this work is not appropriately cited as reference 49. Very recently, the first NHC-catalyzed atroposelective synthesis of N–N axially chiral compounds has been reported by Biju group, which is not cited herein (ChemRxiv, 2023, 10.26434/chemrxiv-2023-7zxsu). Compared with the above-mentioned literature examples, the work herein can be considered as the following-up studies. In short, the closely related works need to be cited appropriately, in a right context – the authors obviously did not do that.

Some minor points: the scope of the reaction is rather narrow, and the usefulness of the product/method was not sufficiently demonstrated.

Reviewer #2 (Remarks to the Author):

In this manuscript, Lu and coworkers reported a direct synthesis of N–N atropisomers with simultaneous creation of contiguous axial and central chirality by oxidative NHC catalyzed [3 + 3] cyclization of isothioureas with enals, the strategy also enabled facile access to N–N axially chiral

indoles from thioureas with ynals. Structurally diverse N–N axially chiral pyrroles and indoles with vicinal central chirality were obtained in moderate to good diastereoselectivities and excellent enantioselectivities. Although this reaction strategy has been used in the C–N axis chiral construction, the reaction synthesizes a series of novel N–N axis chiral molecules with a novel skeleton through this strategy, which is of great significance. Overall, the manuscript can be accepted for publication in Nature Comm. after the following issues are addressed.

1) On line 55 of the second page, the authors indicated that they could synthesize the N–N axis chirality of the indole through 3+3 reaction, but this part was all about the synthetic pyrrole axis chirality, and the authors needed to investigate the applicability of the indole substrate.

2) The product 7d should have two axial chirality, C–N and N–N. The authors need to explain here, give diastereoselectivity results, and explain the reduced enantioselectivity of the reaction.

3) The definition of the pictures in the single crystal part and the calculation part of the paper is too low, so the author needs to change to the pictures with high definition.

4) In the supporting information section, the compound structure formula format is not uniform, and the carbon spectrum of the product should retain one decimal number.

5) In Table 1, the authors should give the diastereoselectivity under different conditions, does the diastereoselectivity of the reaction change by changing the base or the solvent.

6) In this paper, many forms are not uniform, including Spaces before and after brackets, which the author needs to modify carefully. Also in fig 5, scheme 6 should be changed to figure 6.

7) In Table 8, the reactant of eq.3, alkyne is omitted.

8) In the gram preparation experiment, can the amount of catalyst for the reaction be reduced?

Reviewer #3 (Remarks to the Author):

This manuscript by Lu, Wong and coworkers reported atroposelective synthesis of N–N axially chiral indoles and pyrroles via NHC-catalyzed asymmetric formal (3+3) cycloadditions of indole or pyrrole-based platform molecules with enals or ynals. More importantly, N–N axially chiral indoles bearing both axial and central chirality were synthesized in high diastereoselectivity with simultaneously controlling the multiple chiral elements. This approach has a wide substrate scope, and this class of axially chiral compounds has high rotational barrier and configurational stability. Moreover, the authors performed DFT calculations to reveal the origins of the diastereoselectivity and the role of the substituents on the N-phenyl ring of the triazolium catalyst. This work has some important features:

(1) This work has realized the first catalytic asymmetric N–N axially chiral indoles bearing both axial and central chirality. In recent years, the construction of N–N axially chiral skeletons and the

construction of axially chiral indole-based frameworks have become emerging research areas with great significance. Particularly, simultaneously controlling the multiple chiral elements (axial chirality and central chirality) in a high diastereo- and enantioselective manner is very challenging. This work has solved this great challenge and accomplished high diastereo- and enantioselective synthesis of N–N axially chiral indoles bearing both axial and central chirality, which is a breakthrough in the related fields.

(2) This work has provided a powerful strategy for constructing N–N axially chiral indole and pyrrole-based scaffolds. The authors designed and synthesized indole and pyrrole-based isothioureas and thioureas as competent platform molecules, which can readily undergo formal (3+3) cycloadditions under NHC catalysis. In fact, this class of isothioureas and thioureas platform molecules are promising for undergoing other useful transformations, which will open an avenue for constructing different types of N–N axially chiral indole and pyrrole-based scaffolds.

(3) This work has offered an in-depth understanding of the origins of the diastereoselectivity and the role of the NHC catalyst. By theoretical investigations, the authors have disclosed the reason for the observed high diastereoselectivity and the activation mode of chiral NHC catalyst to the substrates, which will enlighten the design of the strategy for controlling the multiple chiral elements in the related transformations.

Therefore, this work has high originality, novelty and significance, which will absorb intense and wide interests of scientists from heterogeneous research areas such as synthetic chemistry, asymmetric catalysis, medicinal chemistry and materials science. For the above features and reasons, this work is strongly recommended for publication in Nature Communications after addressing some minor issues as follows.

(1) In the title and other places, it is suggested to change “cyclization” into “(3+3) cycloaddition” because “cyclization” commonly denotes an intramolecular reaction. In addition, it is suggested that the authors briefly mention that organocatalytic asymmetric (3+3) cycloaddition is a powerful method for constructing six-membered rings, which might trigger them to design this powerful strategy.

(2) In the introduction, it is suggested that the authors mention that the construction of axially chiral indole-based frameworks has become an important research area with the citation of a recent review (Acc. Chem. Res. 2022, 55, 2562). In addition, they should also mention that simultaneously controlling the multiple chiral elements (axial chirality and central chirality) in a high diastereo- and enantioselective manner is very challenging with the citation of a recent review (Acc. Chem. Res. 2022, 55, 2545) and recent publications (For example: Fundamental Research, 2023, 3, 237; J. Org. Chem. 2023, DOI: 10.1021/acs.joc.2c02303).

(3) In the caption of Fig. 6 and other places, it is suggested to change “N–N axially chiral pyrroles with 2,3-dihydropyrimidin-4-ones” into “N–N axially chiral pyrroles bearing an 2,3-dihydropyrimidin-4-one moiety”.

(4) The supplementary materials are in high quality. But there are some minor issues to be corrected.

In page S5, the scheme should be drawn in the format of ACS 1996.

In page S6, the range of melting point for compound 1n is too large, which should be re-checked.

In page S16, in the general procedure, the authors mention the use of glovebox. Is it necessary to use glovebox for this organocatalytic reaction? Have the authors tried the reaction without the use of glovebox? In addition, they mentioned "The vial was then sealed and the reaction mixture was allowed to stir at 30 oC for overnight." But it is not clear whether this step should also be performed in glovebox. Please clarify this issue.

In section "7. Characterizations of N–N axially chiral pyrroles and indoles", the absolute configuration of axial chirality should be described as "Ra" or "Sa" (a should be subscript).

Reviewer #4 (Remarks to the Author):

In this work, the authors report a direct catalytic synthesis of N–N atropisomers with simultaneous creation of contiguous axial and central chirality by oxidative NHC catalyzed [3 + 3] cyclization. Overall, I found this research interesting and well-written. I support its publication in Nature Communications if the following minor issues are properly addressed.

1. I recommend carrying out DFT calculations to understand the origins of enantioselectivity and provide a chiral induction model. This would add more value to the study and enhance its relevance to the field.
2. NLE experiments would be helpful to confirm the proposed mechanistic model.
3. I am concerned that no sufficient information on the computational details is provided in the manuscript or SI. It is essential to ensure that the calculations are reliable, especially regarding the conformational search. Therefore, I urge the authors to provide the detailed information on this aspect, including the structures and energies of all the calculated conformers.

西北工业大学
NORTHWESTERN POLYTECHNICAL UNIVERSITY

Dr. Shenci Lu

Frontiers Science Center for Flexible Electronics,
Shaanxi Institute of Flexible Electronics & Shaanxi
Institute of Biomedical Materials and Engineering
Northwestern Polytechnical University
127 West Youyi Road
Xi'an 710072, P. R. China
E-mail: iamsclu@nwpu.edu.cn

July 28, 2023

Thank you very much for handling our manuscript entitled “**Diastereo- and Atroposelective Construction of N–N Axially Chiral Pyrroles and Indoles through an Organocatalytic Cyclization**” (Manuscript number: NCOMMS-23-06331). We sincerely appreciate the valuable comments from the reviewers. We have managed to address all of them and I am happy to submit revised manuscript and supporting information for your consideration. Please find the point-to-point response below.

Revision from referee 1:

1) With regard to the novelty of this work: (a) N–N axially chiral chemistry: the organo-catalyzed atroposelective formation of the N–N bond with pyrrole/indole scaffold has been reported by many research groups (refs 9-18). I do not think that the authors make any conceptual contributions in this submission. (b) NHC chemistry: NHC-catalyzed selective addition of isothioureas to enals to access 5,6-dihydropyrimidin-4-ones has been reported by the Chi group, however that pioneering work is not cited in this manuscript (Org. Chem. Front., 2021, 8, 743). NHC-catalyzed atroposelective cycloaddition reaction between thioureas and ynals has also been developed by the Jin group, however this work is not appropriately cited as reference 49. Very recently, the first NHC-catalyzed atroposelective synthesis of N–N axially chiral compounds has been reported by Biju group, which is not cited herein (ChemRxiv, 2023, 10.26434/chemrxiv-2023-7zxsu). Compared with the above-mentioned literature examples, the work herein can be considered as the following-up studies. In short, the closely related works need to be cited appropriately, in a right context – the authors obviously did not do that.

Response: We appreciate this comment. (a) Indeed both the metal- and organo-catalyzed atroposelective formation of the N–N bond with pyrrole/indole scaffold has been reported by many research groups, as we cited in refs 9-18 in our previous manuscript. However, the use of NHC-catalyzed processes for the construction of chiral N–N axis remained elusive before our work. In addition, the compounds delivered in our studies are structurally distinct from all the previous reports

on N-N axially chiral compounds (containing a special dihydropyrimidin-4-one scaffold with or without a stereogenic center). This certainly contributes to the structural diversity of N–N linked axially chiral compounds.

It is also noteworthy that the construction of N–N axially chiral compounds is attracting great momentum in recent years. Even during the revision of our work, the preparation of indole-pyrrole or bis-pyrroles bearing a N–N chiral axis through well-established Pd-, Ir- or Rh-catalyzed C–H functionalization were reported in *Angew. Chem.* on 20 March 2023 and 04 May 2023, *Chem. Sci.* on 17 July 2023, respectively (*Angew. Chem. Int. Ed.* **2023**, *62*, e202218871; *Angew. Chem. Int. Ed.* **2023**, *62*, e202305067; *Chem. Sci.* **2023**, *14*, DOI: 10.1039/D3SC02800C). In this revision, we have added these references as the new ref 15, 17 and 19. We are certain that our chemistry will attract much interest in the synthetic community.

(b) The Chi group's work about NHC-catalyzed selective addition of isothiourreas to enals to access 5,6-dihydropyrimidin-4-ones (*Org. Chem. Front.*, 2021, *8*, 743) has been added as the new ref 59 in the revised manuscript. We have also adjusted the position of the previous reference 49 (Jin's work), and put it in the appropriate place in the revised manuscript as ref 64. These important reaction developments have been properly discussed in the revised manuscript.

(c) It is important to note that our work was submitted to *Nat. Commun.* on 11 February 2023. After submission of our work, the preparation of 3-amino Quinazolinones bearing a N–N chiral axis through an atroposelective N-Acylation appeared in a preprint on 27 February 2023, see: Balanna, K., Barik, S., Barik, S., Shee, S., Manoj, N., Gonnade, R. G. & Biju, A. T. N-Heterocyclic Carbene-Catalyzed Atroposelective Synthesis of N–N Axially Chiral 3-Amino Quinazolinones. <https://doi.org/10.26434/chemrxiv-2023-7zxsx>. It was impossible for us to have cited this work in our original submission. In this revision, we have added this reference as the new ref 51.

2) Some minor points: the scope of the reaction is rather narrow, and the usefulness of the product/method was not sufficiently demonstrated.

Response: We appreciate this comment. In order to expand the scope of the reaction, we managed to include three more examples (**7ae**, **7af** and **7ag** in the revised Fig. 6). These structures include substituted indoles instead of pyrroles in other examples, which expanded the core structure of our product. Furthermore, product **7ag** contain a valuable phosphine oxide group, reduction of which to the triarylphosphine was also demonstrated in our revision. To showcase the potential application of this axially chiral phosphine based on a N–N axis, we have further tested it as a chiral ligand in the palladium-catalyzed enantioselective allylic substitution reaction of 1,3-diphenylallyl acetate **14** and dimethyl malonate **15**. The target product **16** was formed in 81% yield with a moderate 40% ee (Fig. 8, eq. 6). Although the enantioselectivity needs to be further improved, this attempt demonstrated the potential application of the constructed axially chiral indoline bearing a 2,3-dihydropyrimidin-4-one

moiety scaffold for the development of a new type of chiral ligands.

Revision from referee 2:

1) On line 55 of the second page, the authors indicated that they could synthesize the N-N axis chirality of the indole through 3+3 reaction, but this part was all about the synthetic pyrrole axis chirality, and the authors needed to investigate the applicability of the indole substrate.

Response: We appreciate this comment. We managed to include three more examples (**7ae**, **7af** and **7ag**) that demonstrate the compatibility with the indole substrates. The data are included in the updated Fig. 6 with characterization in the revised SI.

2) The product **7d** should have two axial chirality, C–N and N–N. The authors need to explain here, give diastereoselectivity results, and explain the reduced enantioselectivity of the reaction.

Response: We greatly appreciate this comment. In order to further confirm the structure of previous **7d**, an analogous substrate **5ai** bearing bromo group was employed in the reaction, and the structure of product **9b** was unambiguously assigned by single crystal X-ray diffraction analysis. This result showed that this reaction followed a different reaction pathway from all the other examples in pervious Figure 6, and the right structure of the previous **7d** should be **9a**. For this series of products

resulting from conjugate addition of the sulfur unit, lower level of enantioselectivities were obtained for **9a-9c** in the revised manuscript. This type of thiazine derivatives with C–N axial chirality was reported by the Jin’s group (*Angew. Chem. Int. Ed.* **2021**, *60*, 9362–9367. Ref 64). This reaction likely followed the mechanism shown below, which is included in the revised SI (Fig. S3). The compound **7l** (previous **7m**) was confirmed by single crystal X-ray diffraction analysis in the original manuscript. The compounds (**7a-7k**, **7m-7ag**) in the revised manuscript were rechecked, and they are all correct. This result showed that the *para* or *meta* substituted on the N-phenyl group of the thioureas afforded the N–N axially chiral pyrroles and indoles bearing a 2,3-dihydropyrimidin-4-one moiety **7**. The mechanism for the forming **7** is also included in the revised SI (Fig. S1).

3) The definition of the pictures in the single crystal part and the calculation part of the paper is too low, so the author needs to change to the pictures with high definition.

Response: We appreciate this comment. We have changed the pictures in the single crystal part and the calculation part with high definition.

4) In the supporting information section, the compound structure formula format is not uniform, and the carbon spectrum of the product should retain one decimal number.

Response: We appreciate this suggestion. We redrew the compound structure formula in the format of ACS 1996. We also handled the carbon spectrum of the product which retain one decimal number.

5) In Table 1, the authors should give the diastereoselectivity under different conditions, does the diastereoselectivity of the reaction change by changing the base or the solvent.

Response: We appreciate this comment. The diastereoselectivity data were added in different conditions in Table 1.

6) In this paper, many forms are not uniform, including Spaces before and after brackets, which the author needs to modify carefully. Also in fig 5, scheme 6 should be changed to figure 6.

Response: We appreciate this important insight. We have rechecked the forms carefully and modified them. In addition, in Fig 5, scheme 6 has been changed to Fig 6.

7) In Table 8, the reactant of eq.3, alkyne is omitted.

Response: We appreciate this important insight. The reactant of alkyne (Ethynyltrimethylsilane) was added in Fig 8. Eq .3.

8) In the gram preparation experiment, can the amount of catalyst for the reaction be reduced?

Response: We appreciate this comment. When the catalyst loading was 20 mol%, the reaction provided the desired product **3a** in 72% yield with 92% ee. When the catalyst was reduced to 10 mol%, the reaction afforded the desired product **3a** in much lower yield (33%) with 88% ee.

Revision from referee 3:

1) In the title and other places, it is suggested to change “cyclization” into “(3+3) cycloaddition” because “cyclization” commonly denotes an intramolecular reaction. In addition, it is suggested that the authors briefly mention that organocatalytic asymmetric (3+3) cycloaddition is a powerful method for constructing six-membered rings, which might trigger them to design this powerful

strategy.

Response: We appreciate this suggestion. We have changed “cyclization” into “(3+3) cycloaddition” in the revised manuscript. We added the sentence “in the construction of six-membered rings through asymmetric (3 + 3) cycloaddition using α,β -unsaturated acylazoliums as C3 synthon.”

2) In the introduction, it is suggested that the authors mention that the construction of axially chiral indole-based frameworks has become an important research area with the citation of a recent review (Acc. Chem. Res. 2022, 55, 2562). In addition, they should also mention that simultaneously controlling the multiple chiral elements (axial chirality and central chirality) in a high diastereo- and enantioselective manner is very challenging with the citation of a recent review (Acc. Chem. Res. 2022, 55, 2545) and recent publications (For example: *Fundamental Research*, 2023, 3, 237; *J. Org. Chem.* 2023, DOI: 10.1021/acs.joc.2c02303).

Response: We appreciate this comment. We added the sentence “Particularly, atropisomeric indole derivatives play an important role in discovering and developing pharmaceuticals.^{3,4} Therefore, the construction of axially chiral indole-based frameworks has become an important research.⁵” and the new ref 5 (Acc. Chem. Res. 2022, 55, 2562). In addition, we added the sentence “Recently, simultaneously controlling multiple chiral elements (axial chirality and central chirality) has emerged as an important research area with several pioneering works reported in the past decade.²⁶⁻²⁸” and the new ref 26-28 (Acc. Chem. Res. 2022, 55, 2545; *Fundamental Research* 2023, 3, 237 and *J. Org. Chem.* 2023, 88, 7684–7702.)

3) In the caption of Fig. 6 and other places, it is suggested to change “N–N axially chiral pyrroles with 2,3-dihydropyrimidin-4-ones” into “N–N axially chiral pyrroles bearing a 2,3-dihydropyrimidin-4-one moiety”.

Response: We appreciate this suggestion. We have changed “N–N axially chiral pyrroles with 2,3-dihydropyrimidin-4-ones” to “N–N axially chiral pyrroles and indoles bearing a 2,3-dihydropyrimidin-4-one moiety” in the revised manuscript.

4) The supplementary materials are in high quality. But there are some minor issues to be corrected.

a) In page S5, the scheme should be drawn in the format of ACS 1996.

Response: We appreciate this comment and redrew the scheme in page S5 in the format of ACS 1996.

b) In page S6, the range of melting point for compound **1n** is too large, which should be re-checked.

Response: We appreciate this comment and tested the melting point for compound **1n** again.

c) In page S16, in the general procedure, the authors mention the use of glovebox. Is it necessary to use glovebox for this organocatalytic reaction? Have the authors tried the reaction without the use of glovebox? In addition, they mentioned “The vial was then sealed and the reaction mixture was allowed to stir at 30 °C for overnight.” But it is not clear whether this step should also be performed in glovebox. Please clarify this issue.

Response: We appreciate this comment. It is not necessary to use glovebox for this organocatalytic reaction. We tried this reaction with the use of Schlenk tube under an atmosphere of nitrogen. We have also noted the difference in the yield of **3a** and **7a** with the use of Schlenk tube or glovebox. This result showed that the reaction with the use of Schlenk tube afforded slight lower yield with the same enantioselectivity.

“The vial was then sealed and the reaction mixture was allowed to stir at 30 °C for overnight.” It is not necessary to perform this step reaction in glovebox. We have modified the description of the general procedure shown in page S16 in the revised supporting information. “The reaction mixture was taken outside the glovebox. The vial was then sealed and the reaction mixture was allowed to stir in the fume hood at 30 °C for overnight.”

d) In section “7. Characterizations of N–N axially chiral pyrroles and indoles”, the absolute configuration of axial chirality should be described as “*R_a*” or “*S_a*”.

Response: We appreciate this suggestion. In section “7. Characterizations of N–N axially chiral pyrroles and indoles”, the absolute configuration of axial chirality has been described as “*R_a*” or “*S_a*”.

Revision from referee 4:

1) I recommend carrying out DFT calculations to understand the origins of enantioselectivity and provide a chiral induction model. This would add more value to the study and enhance its relevance to the field.

Response: We appreciate this suggestion. With the new mechanistic insights from experimental

study of NLE effect, we were able to revise the proposed mechanism and calculate the origins of enantioselectivity and diastereoselectivity using the new a transition state model involving two NHC catalysts molecules (See below discussion for NLE experiments). Unfortunately, with the increased size of the system and a weak linkage between the second catalyst and isothiourea, quantitative agreement with experimental selectivities was not achieved and we were only able to get qualitative agreement (Table S2). The chiral induction is consistent with the transition state model we reported earlier (*Chem. Eur. J.* **2017**, *23*, 2275-2281).

2) NLE experiments would be helpful to confirm the proposed mechanistic model.

Response: We greatly appreciate this suggestion. Indeed, the result of NLE experiments revealed new mechanistic insights into the reaction. We observed a negative non-linear effect, which points to the involvement of a second catalyst molecule in the enantio-determining step. We further substantiated it by NMR experiments, which support the existence of non-covalent interaction between NHC catalyst and isothiourea. This interaction was proposed to activate isothiourea towards nucleophilic addition reaction. We revised our proposed mechanism and re-did our calculations accordingly. We believe the NLE experiments have greatly enhanced the credibility of the proposed mechanism.

3) I am concerned that no sufficient information on the computational details is provided in the manuscript or SI. It is essential to ensure that the calculations are reliable, especially regarding the conformational search. Therefore, I urge the authors to provide the detailed information on this aspect, including the structures and energies of all the calculated conformers.

Response: We appreciate this suggestion. Computational details, including a computational methods section, a section on the study of enantioselectivity and diastereoselectivity, and the complete set of atomic coordinates and their thermodynamics data, have been added in supporting information. We agree that conformation search is essential for reliable computational results, especially for flexible systems with many degrees of freedom. In the current case, due to the high rigidity of both catalysts and substrates, our search yielded only a few distinct conformers. We have included higher-energy conformers whenever possible.

Thank you for your kind consideration. I am looking forward to hearing from you.

Best regards,

Shenci Lu

Shenci Lu

REVIEWER COMMENTS

Reviewer #2 (Remarks to the Author):

I appreciate the authors' effort to improve the manuscript and the authors have addressed all my concerns.

Reviewer #4 (Remarks to the Author):

In the revised manuscript, the authors discovered that a NLE effect, which certainly require the involvement of secondary chiral NHC for the enantioselectivity control. However, the DFT and mechanistic rationale are incomplete. The claimed stereoselectivity-determining F-TS2 have no correspondence for the observed enantioselectivity, and I don't think the size of the computational system justifies the correctness of the proposed model. Therefore, I don't support the publication of this work in its current version.

I think the authors need to perform thorough DFT mechanistic study to clarify the origin of enantioselectivity, otherwise the current experimental and computational results are not consistent.

Dear referees,

Thanks for your valuable and expert comments on our work very much. We carefully revised our manuscript and provided our revisions and responses to your comments by point-to-point in this letter.

Revision from referee 2:

I appreciate the authors' effort to improve the manuscript and the authors have addressed all my concerns.

Response: Thank you very much. It's our honor to receive your positive comment.

Revision from referee 4:

In the revised manuscript, the authors discovered that a NLE effect, which certainly requires the involvement of secondary chiral NHC for the enantioselectivity control. However, the DFT and mechanistic rationale are incomplete. The claimed stereoselectivity-determining F-TS2 have no correspondence for the observed enantioselectivity, and I don't think the size of the computational system justifies the correctness of the proposed model. Therefore, I don't support the publication of this work in its current version.

I think the authors need to perform thorough DFT mechanistic study to clarify the origin of enantioselectivity, otherwise the current experimental and computational results are not consistent.

Response: We appreciate this comment. We agree that the best confirmation of the mechanistic proposal that the stereoselectivity-determining TS is **F-TS2** is to predict the experimental enantioselectivity of the reaction, in addition to diastereoselectivity. In this revised manuscript, we carried out a more thorough computational study about the enantioselectivity of the reaction. We constructed and optimized 10 more conformers of **F-TS2-SS_a**, the lowest-energy conformer of which is 18.6 kJ/mol higher than the lowest-energy **F-TS2-RR_a**. This represents a 3.8 kJ/mol lowering of the energy of **F-TS2-SS_a** reported in earlier version of the manuscript. Despite this improvement, the energy is still 10.9 kJ/mol higher than expected by the experimental ee of 91% at 30 °C, which corresponds to an energy difference of 7.7 kJ/mol.

Conformation	H _{corr}	G _{corr}	E _{6-31G(d)}	E _{6-311+G(2d,p)}	ΔΔH [‡]	ΔΔG [‡]	ΔΔG [‡] -ΔΔH [‡]
F-TS2-RR_a	1.588453	1.371832	-4411.116648	-4412.346341	0.0	0.0	0.0
F-TS2-SS_a-CONF1	1.588626	1.369764	-4411.104671	-4412.335726	28.3	22.4	-5.9
F-TS2-SS_a-CONF2	1.588773	1.371227	-4411.105794	-4412.334957	30.7	28.3	-2.4
F-TS2-SS_a-CONF3	1.588413	1.371225	-4411.109326	-4412.338650	20.1	18.6	-1.5
F-TS2-SS_a-CONF4	1.588401	1.372153	-4411.107596	-4412.336676	25.2	26.2	1.0
F-TS2-SS_a-CONF5	1.588928	1.374450	-4411.108439	-4412.337995	23.2	28.8	5.6
F-TS2-SS_a-CONF6	1.589151	1.376382	-4411.110351	-4412.339692	19.3	29.4	10.1
F-TS2-SS_a-CONF7	1.588795	1.372637	-4411.104434	-4412.334229	32.7	33.9	1.2
F-TS2-SS_a-CONF8	1.588580	1.371575	-4411.103937	-4412.332645	36.3	35.3	-1.0
F-TS2-SS_a-CONF9	1.588245	1.371564	-4411.099491	-4412.332253	36.4	36.3	-0.2
F-TS2-SS_a-CONF10	1.588499	1.374319	-4411.106695	-4412.334819	30.4	36.8	6.4
F-TS2-SS_a-CONF11	1.588277	1.374516	-4411.098292	-4412.328203	47.2	54.7	7.5
F-TS2-SS_a-CONF12	1.588278	1.377580	-4411.095142	-4412.321508	64.7	80.3	15.6

The large calculation error can be attributed to two possible sources. The first is the challenge of accurately calculating the entropic contribution to energy difference, as can be seen from the wide fluctuation of the $\Delta\Delta G^\ddagger - \Delta\Delta H^\ddagger$ term in the last column of the table. This is especially difficult to handle when there are many very low-lying frequencies, which is unfortunately the case with a weakly bounded second NHC catalyst. It's important to note that the challenge is universal to computational chemistry, not only to our calculations.

The second possible source is related to the challenge of conformational search of **F-TS2-SS_a**. Despite our attempt to study more conformers, it's very challenging to come up with a systematic approach to the problem. Unlike the origin of diastereoselectivity, which relies on the differentiation of an internal N-N axis with the second NHC catalyst staying more or less at the same place, the origin of enantioselectivity relies on the second catalyst staying at different locations and interacting differently with the rest of the TS. However, the second NHC catalyst interacts only through a weak non-covalent interaction, opening many possibilities to its location and orientation. Our earlier developed program QMTSDock (<https://doi.org/10.1021/acs.jpca.9b09543>) potentially can handle the conformation search of TS automatically, but only at a very low level of method, due to the large size of the system.

In summary, quantitative agreement between theory and experiment is the best way to confirm our mechanistic proposal of the stereoselectivity-determining TS, but it is difficult to achieve in this case. Furthermore, even experimentally confirmed mechanism could latter on be shown to need revision (Wittig Rearrangement is a good example). With the better agreement with experimental ee, we believe the manuscript is ready for publication without unnecessarily delay of the publication of the more important experimental results.

The Revised Supporting Information with all the changes highlighted with a yellow background is included for your easy reference.

Sincerely yours,

Best wishes,

Dr. Lu

REVIEWERS' COMMENTS

Reviewer #4 (Remarks to the Author):

The authors have partially addressed my previous question. Although their computations are not in full agreement with the experimental stereoselectivity, and there still lacks the mechanistic explanations of the negative NLE, I guess the current version is OK if the authors keep stressing it.

Dear referee,

Thanks for your valuable and expert comments on our work very much. We carefully revised our manuscript and provided our revisions and responses to your comment by point-to-point in this letter.

Revision from referee 4:

The authors have partially addressed my previous question. Although their computations are not in full agreement with the experimental stereoselectivity, and there still lacks the mechanistic explanations of the negative NLE, I guess the current version is OK if the authors keep stressing it.

Response: We appreciate this comment. The sentence “Nevertheless, there seemed a qualitatively agreement between our calculations and experiments” has been changed to “It should be emphasized that our calculations are not in full agreement with the experimental stereoselectivity and there still lacks a clear mechanistic explanation of the negative NLE.” in the revised manuscript.

The revised manuscript with all the changes highlighted with a yellow background is included for your easy reference.

Sincerely yours,

Best wishes,

Dr. Lu